# Fabrication and Characterization of TiO₂ Coatings on 304 Stainless-Steel Substrate for Efficient Oil/Water Separation

Jose Sico [1], Benjamin Tang [1], Dayana Flores [1], Roy Mouawad [1], Rheyana Punsalan [1], Yong X. Gan [2]
and Mingheng Li [1,*]

1   Department of Chemical and Materials Engineering, California State Polytechnic University,
    Pomona, CA 91768, USA; jlsico@cpp.edu (J.S.); btang@cpp.edu (B.T.); dayanaflores@cpp.edu (D.F.);
    rsmouawad@cpp.edu (R.M.); rjpunsalan@cpp.edu (R.P.)
2   Department of Mechanical Engineering, California State Polytechnic University, Pomona, CA 91768, USA;
    yxgan@cpp.edu
*   Correspondence: minghengli@cpp.edu; Tel.: +1-909-869-3668

**Abstract:** Oil spill accidents have been a prevalent threat to the environment. To aid in clean-up efforts, a stainless-steel filter with a hydrophilic and oleophobic coating was fabricated for efficient and affordable oil/water separation. Two solutions were used to deposit the coatings. One was sourced from a titanium (IV) isopropoxide (TTIP) precursor dissolved into 1-butanol and the other through the mixing of titanium dioxide nanopowder with glacial acetic acid. The solutions were applied to 304 stainless-steel mesh filters of varying aperture sizes ranging from 30 microns to 240 microns. The coating was applied through a multiphase deposition method followed by sintering at 450 °C. The filter performance was evaluated by contact angle measurement and a filtration test using a mixture of motor oil and water, while the surface morphology and structure of the coatings were characterized by SEM-EDS and XRD. The mesh with smaller aperture size showed oil retention improvement of up to 99%. The TiO₂ nanopowder coating, with a 92% oil retention efficiency, outperformed the coating via the TTIP precursor.

**Keywords:** titanium (IV) isopropoxide; titanium dioxide nanopowder; filter; oil/water separation; solution immersion; hydrophilic and oleophobic coating

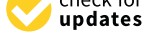



## 1. Introduction

As the global reliance on petroleum has severely increased, its transportation, storage, and extraction have impacted the social, economic, and environmental aspects of sustainability. Oil/water pollution is a continuous problem today, and just the United States alone has been affected by 44 major oil spills, each involving over 10,000 barrels (420,000 gallons), between 1969 and 2017 [1]. One recent oil spill occurred in Southern California in 2021; 25,000 gallons of oil was released into the ocean, harming more than 124 animals [2]. This devastating example was the main inspiration behind this study. Nowadays, there are traditional oil cleanup methods, such as skimming, in situ burning, and dispersing [3]. Skimming is defined as the process that removes oil from the sea surface before it reaches sensitive areas. The devices used in this process attract the oil to the surface and pick up the solution, which is a combination of oil and water [3]. In situ burning is the act of burning spilled oil on the ocean surface [3]. Black smoke and poly-aromatic hydrocarbons (PAHs), both of which being pollutants of concern, are produced as byproducts of in situ burning [4,5]. It was noted that PAHs with greater ring numbers were more detrimental due to greater bioaccumulation and contributed to the presence of carcinogens within the residues. Dispersing, on the other hand, breaks down the oil into smaller droplets by releasing chemical dispersants, allowing them to mix more easily into the water [3]. It is important to note that the effectiveness of chemical dispersants also varies based on the oil-spill conditions. Low temperatures, especially approaching water freezing point,

increase oil viscosity, thereby decreasing the oil-to-dispersant coagulation rate [6]. Each of the methods has their respective deficiencies: poor oil recovery, low separation efficiency, and the risk of introducing additional pollutants into the water.

Multiple oil/water separation techniques, materials, and substrates have been studied and developed by different research groups. Oil/water mixtures can be separated via substrate coatings in two main manners: an oleophilic and hydrophobic property combination, or oleophobic and hydrophilic. Properties matching the former were observed in polydimethylsiloxane (PDMS)/co-polymethylhydrosiloxane (PMHS) coatings [7], $SiO_2$ nanoparticle/dodecyltrimethoxysilane coatings [8], $Si(CH_3)_3$-functionalized $SiO_2$ coatings [9], and copper foils with thiol molecule modifications [10]. Various methods of achieving oleophobic and hydrophilic coating properties were studied as well. In the early 2000s, the first superhydrophobic and superoleophilic material was developed [11]. However, it was found that this substance could be easily contaminated by viscous oil, which significantly decreased its oil separation capabilities [12]. Taking the oil repellency of a fish scale as inspiration, other research groups have developed oleophobic and hydrophilic materials for oil/water separation applications [13]. Past research on this basis includes polyacrylamide hydrogel-coated mesh [14], perfluorosilane-rendered $TiO_2$/single-walled carbon nanotubes [15], and perfluorooctanoic acid-deposited $TiO_2$ [16]. Although most of these materials have shown promising outcomes for the separation of oil/water, not operating within a pre-existing water environment can drastically affect their functionality [12]. Other individuals have focused on using Pickering emulsions coupled with underwater superoleophobic ultrafiltration membranes for oil/water separation [17]; however, the total energy cost needed for the fabrication of the membranes is high.

In 2022, a cost-effective solution to clean oil spills was proposed, and this consisted of modifying a cotton cloth or paper coffee filter with a hydrophilic and oleophobic coating for affordable and efficient oil separation and recovery from water [18]. In the development and fabrication of these coated filters, different techniques were used in adhering $TiO_2$ [18]. It was concluded that the oil/water separation capability was enhanced in all the substrates even though the titanium dioxide present on the surface of the filter was amorphous phase [18].

Even though the outcome of this research was promising, the tested substrates serve mainly as single-use substrates due to their short-term stability. Cloth-, paper-, and cotton-based substrates are limited in reusability and are more difficult to scale for ocean-sized clean-ups [16,18]. Stainless-steel mesh provides a more effective base for large-scale and long-term oil/water separation. Research demonstrates that stainless steel could be treated to last over 50 cycles without a significant reduction in separation efficiency [19]. The corrosion resistance of stainless steel has also been noted as either great or excellent in marine environments [20]. This substrate proves to be an efficient alternative due to its stable composition, long-term usability, low production cost, and varying mesh size ranges. Thus, stainless-steel substrates provide a practical benefit in application when compared to the previously noted substrate materials.

$TiO_2$ was selected as the coating material in this work because it has been widely studied and used in different industries; these include photocatalysts, antibacterial agents, and nano-paints with self-cleaning properties [21]. $TiO_2$ exists in three different crystalline forms, anatase, rutile, and brookite, and it has been shown that when the material has an anatase phase structure, more hydrophilic characteristics can be achieved [22,23]. It was also observed in the 2000s that ultraviolet (UV) illumination could modify the $TiO_2$ surface contact angle to $0°$, making it completely hydrophilic [24]. Anatase $TiO_2$ films have been shown to maintain their hydrophilic properties, with a contact angle less than $20°$, for an extended period of 22 days [25]. Considering the unique surface properties of $TiO_2$, it is used in this study by attaching it onto the stainless-steel meshes. As mentioned, the stainless-steel substrates are well suited for high-temperature processing, during which, the desired anatase crystalline structure can be obtained or retained.

## 2. Materials and Methods

### 2.1. Substrate Materials

The following materials were used as substrates for titania coating: 304 stainless-steel woven wire mesh 80 (aperture size: 0.18 mm), mesh 100 (aperture size: 0.14 mm), mesh 200 (aperture size: 0.077 mm), mesh 300 (aperture size: 0.044), and mesh 400 (aperture size: 0.030 mm).

### 2.1.1. Materials for $TiO_2$ Nanopowder Coating

The following materials were purchased to synthesize the $TiO_2$ nanopowder/nanoparticle coating: glacial acetic acid 99% ACS grade (Lab Alley, Austin, TX, USA) and laboratory-grade titanium dioxide nanopowder from Sigma-Aldrich (St. Louis, MO, USA).

### 2.1.2. Materials for $TiO_2$ Coating from TTIP Precursor

The following materials were involved in the synthesis of $TiO_2$ solution: acetic acid glacial 99% ACS grade (Lab Alley, Austin, TX, USA), titanium (IV) isopropoxide, TTIP (Sigma Aldrich, St. Louis, MO, USA), and 1-butanol (Sigma Aldrich).

### 2.2. Coated Filter Preparation

The solution immersion method was used for its simplicity and the application of consistent layers of the ceramic coating to the substrates [26]. The substates were cut into 1-inch squares and 1-inch radius circles for the purpose of this research. After cutting, they were sanitized in isopropyl alcohol, washed with soapy water, and rinsed with deionized water. After dipping the substrates into the solution, coated substrates were placed in a furnace and sintered at 450 °C for 30 min [27]. Figure 1 depicts a general flowchart for the methods used to prepare the coated filters. Alternatively, the nanopowder–acetic acid mixture can also be applied via an aerosol spray method.

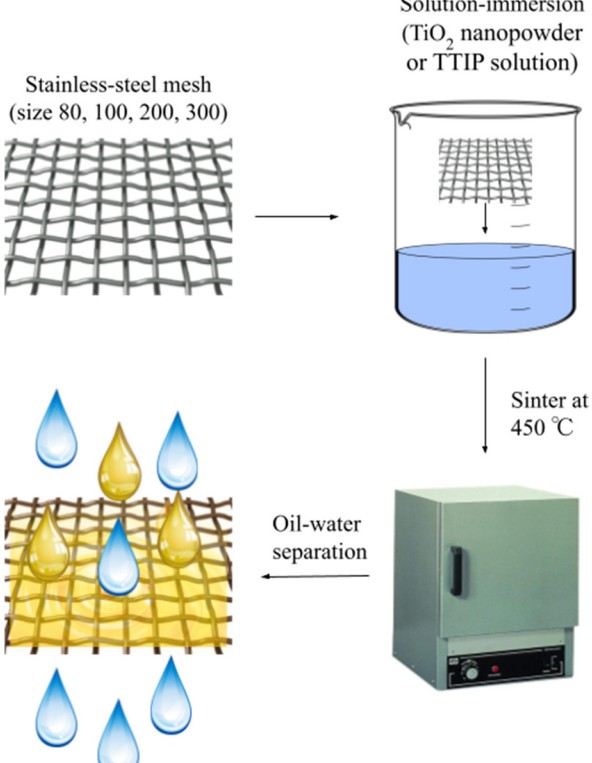

**Figure 1.** Flowchart for coated filter preparation and oil/water separation experiments.

### 2.2.1. Fabrication of TiO$_2$ Nanopowder Coating

TiO$_2$ nanopowder was measured and prepared in a mortar. The solution was prepared by incremental additions of acetic acid to the TiO$_2$ nanopowder in a mortar while gradually grinding with a pestle until a thin paste-like solution was synthesized. The substrates were individually submerged into the solution and sintered in the furnace [27]. When the aerosol spray method was used, the solution was loaded to a spray bottle and then applied to the substrate. Figure 2 shows an example of the sintered coating made via aerosol spraying of a mixture consisting of 5 g of TiO$_2$ nanopowder and 30 mL of acetic acid.

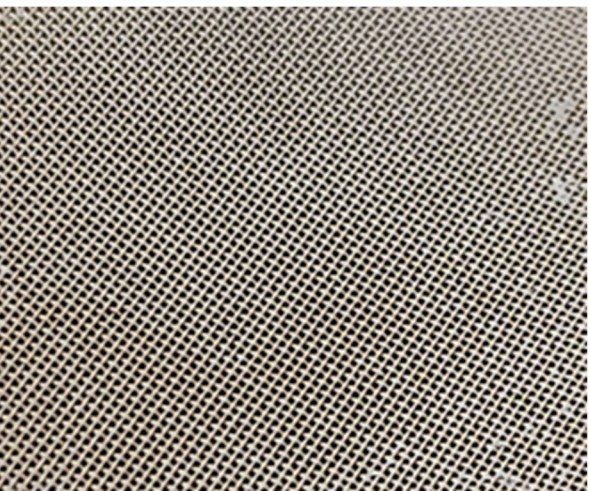

**Figure 2.** Sintered TiO$_2$ coating on stainless-steel substrate.

### 2.2.2. Fabrication of TiO$_2$ Coating from TTIP Precursor

TTIP was dissolved in 1-butanol to form a solution. To create a paste-like solution of the TTIP, the solution was heated for 2 h at 120 °C to boil out the 1-butanol. Incremental additions of glacial acetic acid were added dropwise. Substrates were immersed in the solution when the consistency of the solution was a thin paste. After immersion, the substrates were transferred to another beaker with deionized water to boil out any excess 1-butanol. Once all the 1-butanol was removed, the coated substrates were placed in the furnace.

### 2.3. Performance Testing

Two tests were conducted to evaluate the performance of the coated filter: an oil leakage test and a static contact angle test. The leakage test involves a custom-made filter that can hold the coated substrates in place while preventing any leakage on top of a graduated cylinder. The static contact angle test consists of two methods: the usage of a Kruss K-100 Force Tensiometer and the high-resolution camera of a Samsung Galaxy S23 to capture the frame when the droplet contacts the substrate. The contact angles were determined using ImageJ version 1.46, an image-analyzing software.

### 2.3.1. Oil Leakage Test

A timed gravity leakage test with pure motor oil was conducted to obtain a quantifiable performance of the filtration capabilities and the efficiency of each different mesh size and coating [28]. The test was conducted by pouring 50 mL of Pennzoil Conventional 10W-30 Motor Oil into the funnel, which was then recorded until no more oil was in the original vessel. The whole leakage test was timed and filmed, as shown in Figure 3a. The custom filter vessel was placed above a funnel that was set atop a 50 mL graduated cylinder, as shown in Figure 3b. The time was measured for every mL during the video playback.

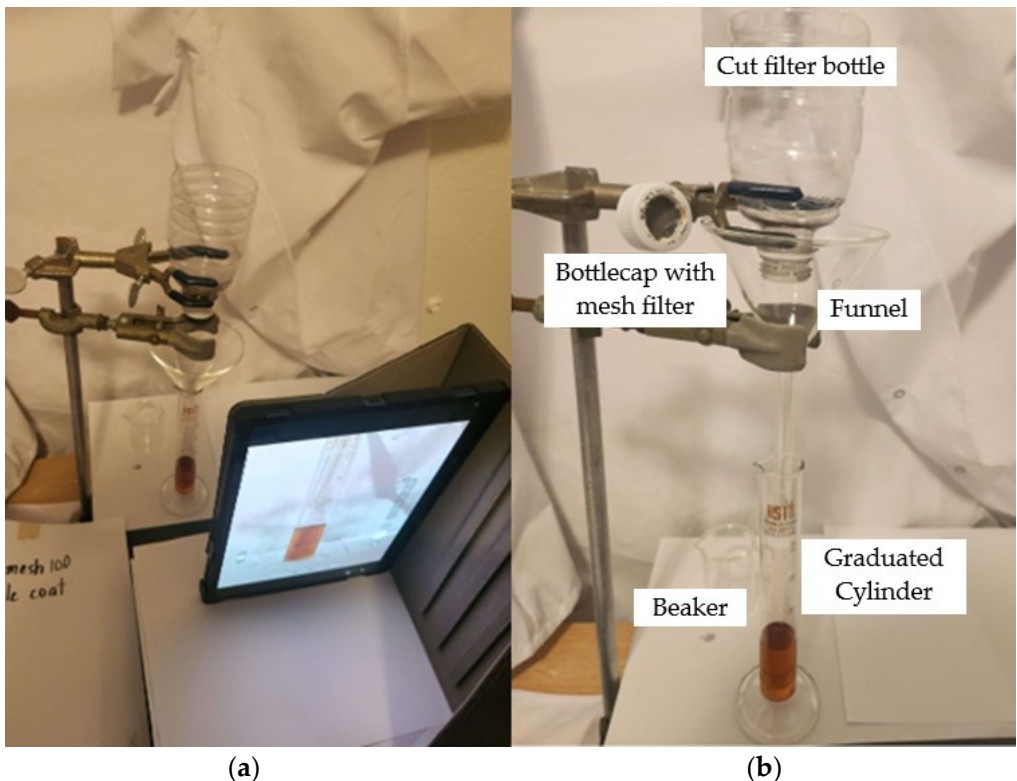

**Figure 3.** (**a**) Overall oil leakage test set-up, with recording device. (**b**) Oil leakage test apparatus with labeling.

The data from the timed filtration was used to compare each substrate and show the performance improvement of the coated filters against control filters. The efficiency of the filter was calculated. The oil retention efficiency $\varepsilon$ was calculated by the difference of the initial volume poured in the filter and the volume collected in the graduated cylinder over a specific length of time divided by the initial volume:

$$\varepsilon = \frac{V_{initial} - V_{filtrate}}{V_{initial}} \times 100\% \tag{1}$$

### 2.3.2. Static Contact Angle Test

The contact angle was used to measure the wettability of the coated filters. A larger contact angle means higher oleophobicity [29,30]. It was hypothesized that the treated filters would have a larger contact angle compared to untreated filters. To measure the contact angle of the filter, a video was taken of a droplet of motor oil (Pennzoil Convention 10W-30) using a syringe. A single frame was taken from the video when the droplet first made contact and was observed with the use of software add-ons (Contact Angle and Drop Shape Analysis) to ImageJ (Version 1.46, U.S. National Institutes of Health, Bethesda, MD, USA). To measure the contact angle, as shown in Figure 4, the averages of the contact angle results from Contact Angle written by Marco Brugnara (for more details, please refer to https://imagej.net/ij/plugins/contact-angle.html (accessed on 11 October 2023)) and Drop Shape Analysis written by Aurélien Stadler and Daniel Sage (https://bigwww.epfl.ch/demo/dropanalysis (accessed on 11 October 2023)) were used.

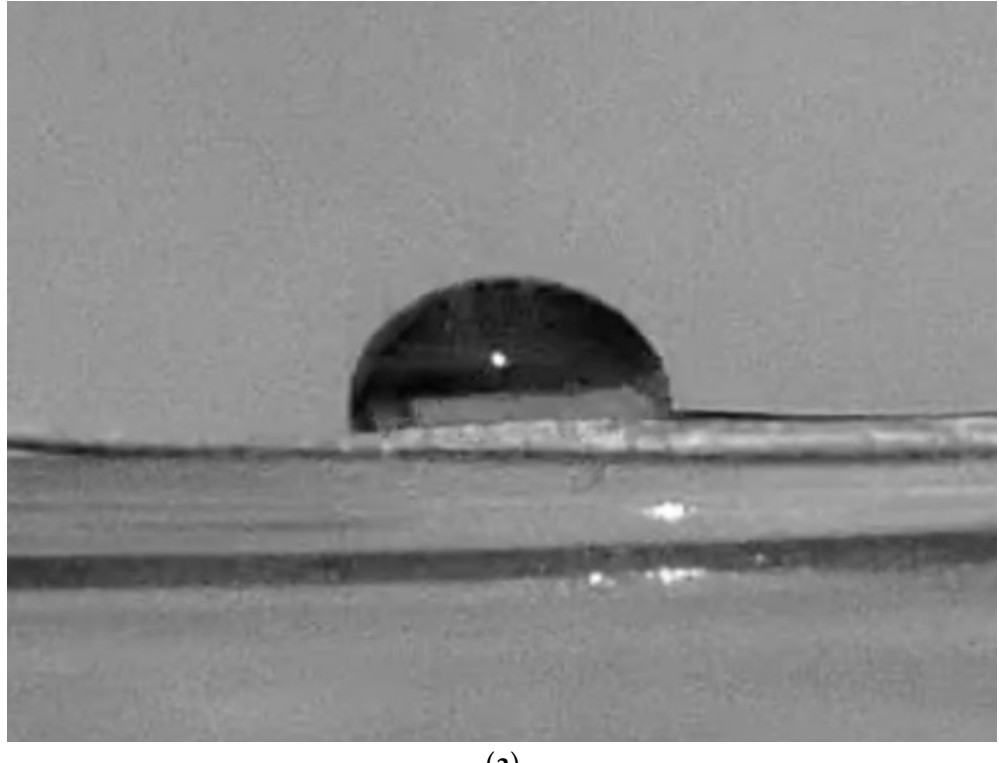

(**a**)

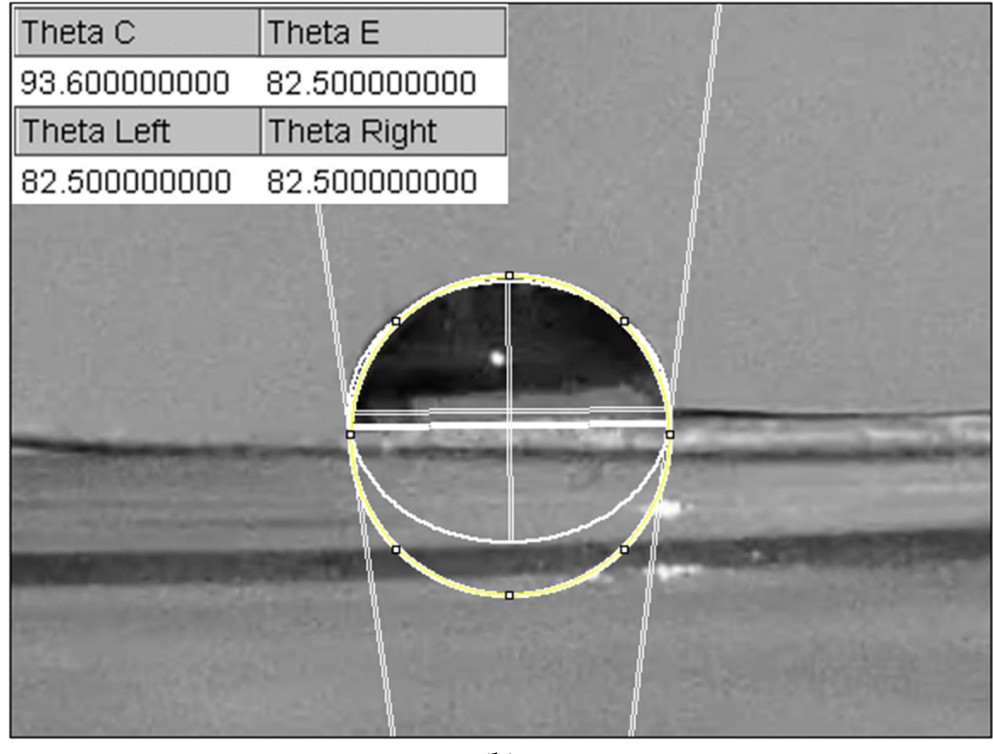

(**b**)

**Figure 4.** *Cont.*

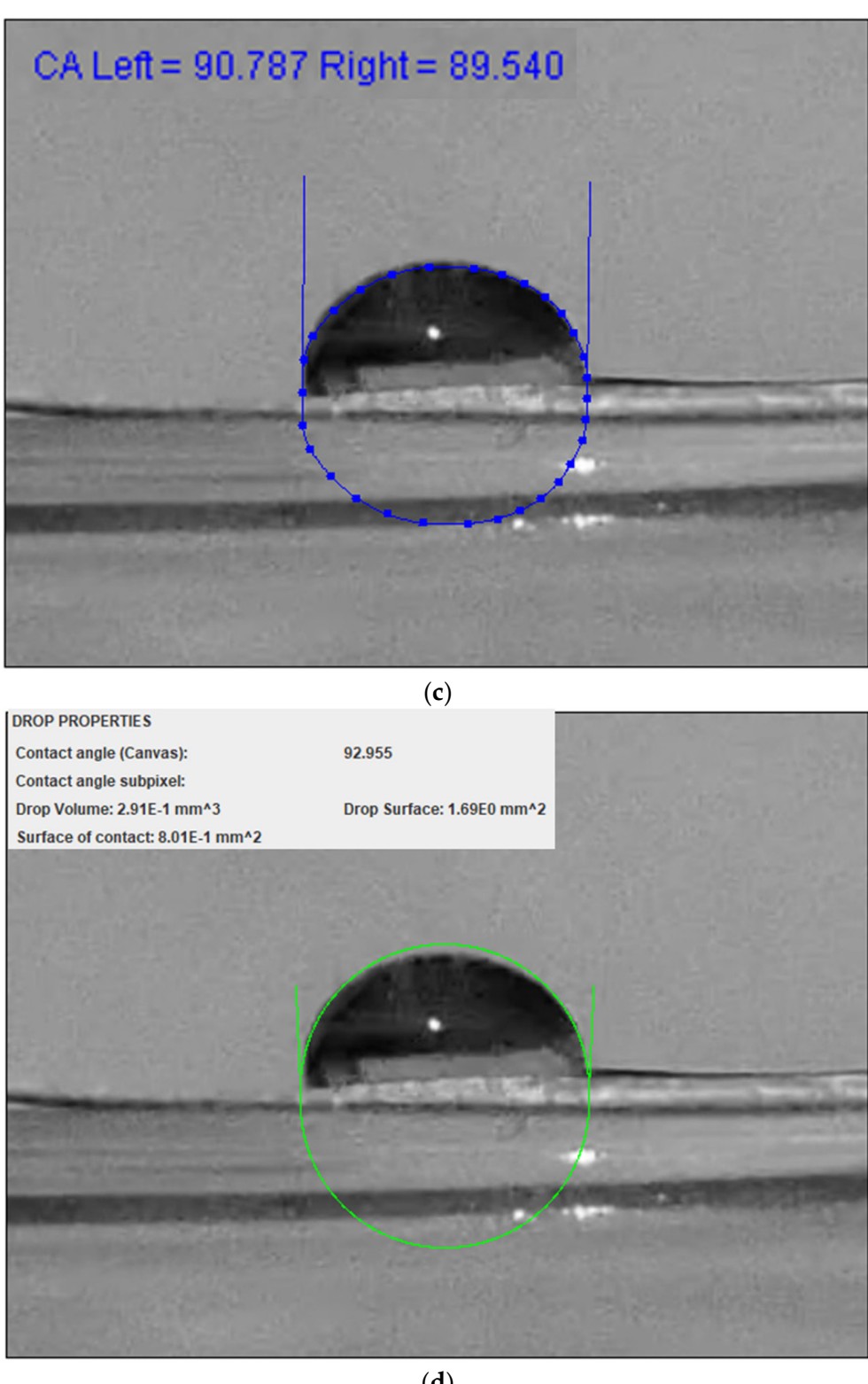

**Figure 4.** ImageJ contact angle analysis of oil droplet on (**a**) TiO$_2$-coated layer of mesh size 300 using (**b**) Contact Angle plugin, (**c**) DropSnake method in the Drop Shape Analysis plugin, and (**d**) Low Bond Axisymmetric Drop Shape Analysis (LB-ADSA) method in the Drop Shape Analysis plugin.

### 2.4. Structure Assessment by SEM, EDS and XRD

A JSM-6010LA scanning electron microscope (SEM) was used to observe the surface of the coating and the substrate. Energy-dispersive X-ray spectroscopy (EDS) was used

to show the element distribution on the specimens. In addition, X-ray diffraction (XRD) was carried out to show the diffraction peaks of the $TiO_2$ coating, and the XRD results were used to confirm the anatase phase structure formation. The XRD instrument model was a Bruker D2 Phaser.

## 3. Results and Discussion

### 3.1. Structure Characterization

The SEM images of the stainless-steel mesh control specimen and the coated filters are shown in Figure 5. The aerosol spray method was applied to obtain a uniform coating on small aperture meshes. Following the aerosol application, the excess liquid solution was removed using compressed air. This process was repeated twice for mesh sizes 100 and 200. Sintering at 450 °C was expected to retain the anatase phase of $TiO_2$, as a comparable temperature (500 °C) has been studied among temperature ranges between 500 to 900 °C [31]. Temperatures greater than 900 °C have been proposed as anatase–rutile phase transition temperatures for metastable $TiO_2$ [32]. Thus, it was expected that the desired hydrophilic properties of anatase $TiO_2$ would be retained throughout the sintering process.

The surface morphology of the samples could clearly be viewed as well. Figure 5b,c show relatively smooth surface coatings with small, embedded crystalline formations. The stainless-steel substrate diameter was estimated to be 73 μm based on ImageJ analysis. Single-layer coating increased the diameter to approximately 79 μm, which translated to a coating thickness of 3 μm. Double-layer coatings provided a predictable increase in thickness; the diameter was approximately 85 μm. This provided an additional 3 μm from the single-layer coat and a total thickness of 6 μm on the stainless-steel substrate.

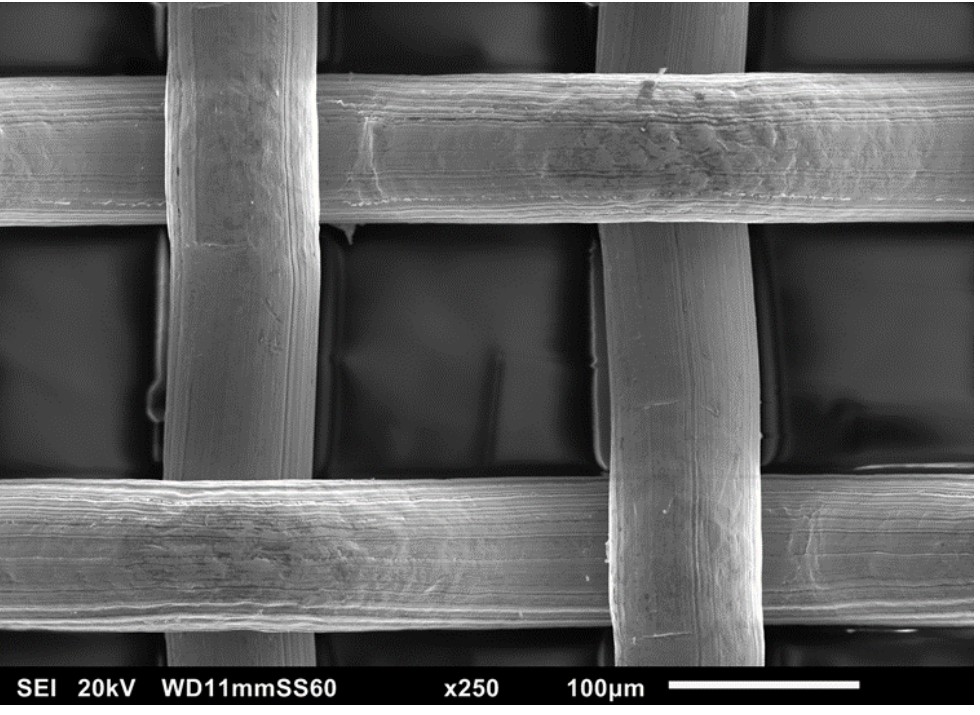

(a)

**Figure 5.** *Cont.*

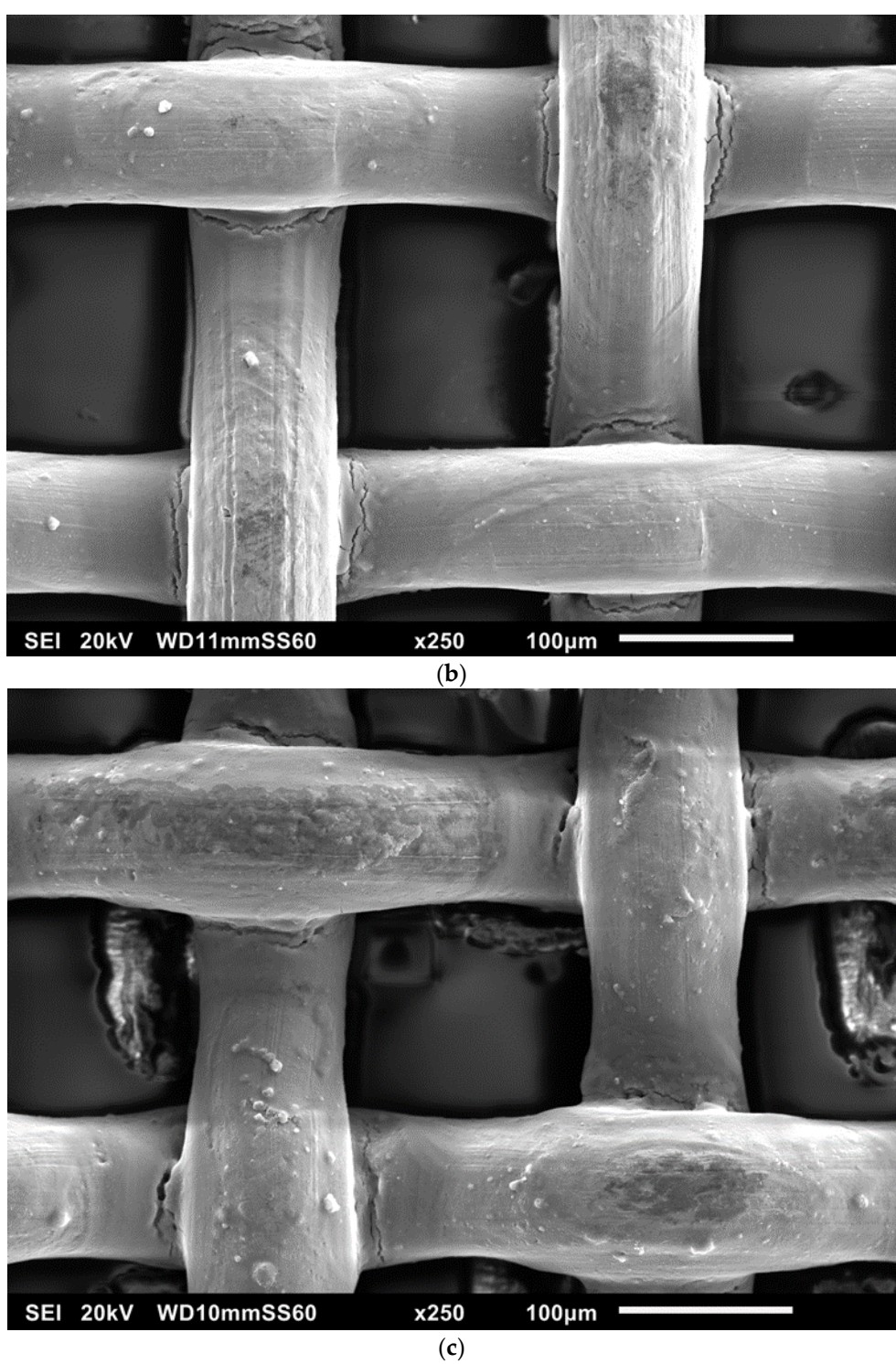

**Figure 5.** SEM image of stainless-steel mesh 100 with conditions: (**a**) uncoated (control); (**b**) single-layer $TiO_2$ nanopowder coating; (**c**) double-layer $TiO_2$ nanopowder coating.

Elemental analysis and structure characterization of the coating were carried out using EDS and XRD. The elemental results from EDS provide insight into the distribution and the relative abundance of elements such as titanium, oxygen, and iron, as shown by the color maps in Figure 6. The titanium presence should be related to the $TiO_2$ present from the nanopowder coating. The atomic proportion of titanium in the single-layer coating was 10.2%. This further increased to 13.5% when an additional layer of coating was applied. XRD was performed with a Bruker D2 Phaser; it has been observed that anatase $TiO_2$

peaks are found predominantly at 2θ values: 24.8°, 37.3°, 47.6°, 53.5°, 55.1°, and 62.2° [33]. Figure 7 depicts the referenced 2θ values for anatase TiO$_2$ with solid, vertical red lines. Discrediting potential noise from the mounting plate and substrate material, the graphical results indicate a clear presence of anatase phase TiO$_2$ from the coating. Strong diffraction peaks at 24.8°and 47.6° provided confirmation of the anatase structure [34].

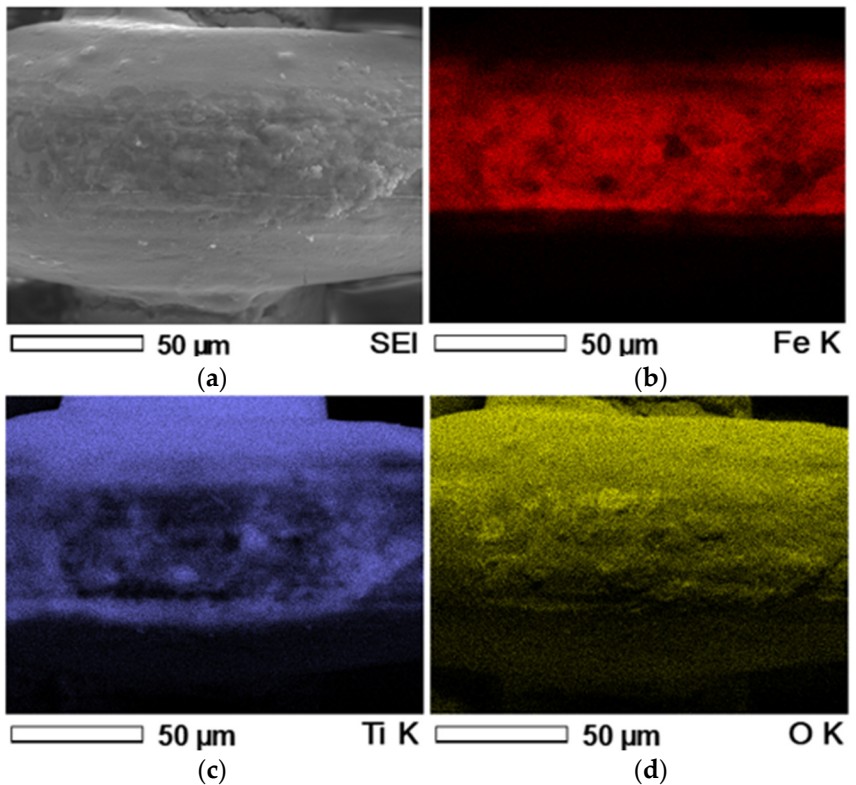

**Figure 6.** EDS of specific elements on two-layer TiO$_2$ nanoparticle coating on mesh 100. (**a**) Raw scan; (**b**) Fe; (**c**) Ti; (**d**) O.

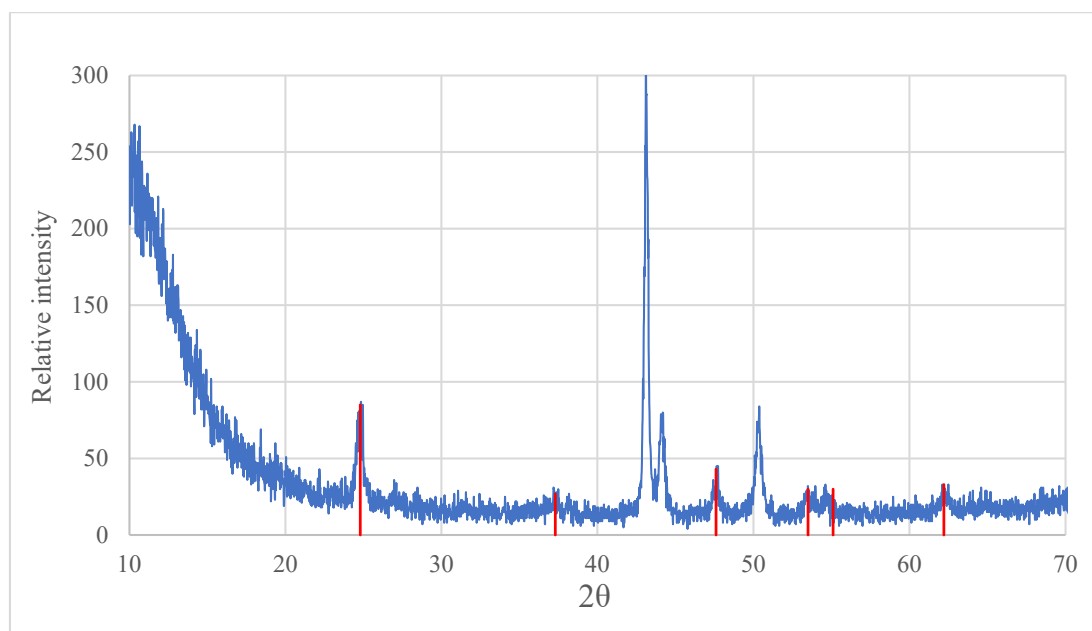

**Figure 7.** XRD results of fabricated TiO$_2$ nanopowder-coated filters, with referenced anatase phase diffraction peaks depicted in red.

### 3.2. Oil Leakage and Oil/Water Separation Efficiency

Figure 8 shows the data collected for coated and control mesh size 100 and mesh size 200. Mesh size 80 had similar results to mesh size 100. The $TiO_2$ coating completely clogged the mesh size 400, and thus, did not allow any fluid to pass through. Consequently, the results from mesh sizes 80 and 400 results were disregarded. The smaller volumes of oil filtrate represent a better separation performance. The results demonstrate that coated meshes show a significant improvement as opposed to the uncoated filters. This was expected as the $TiO_2$ coating composition has significant oleophobic properties. The $TiO_2$ nanopowder-coated filter performed slightly better than the one made from TTIP precursor for both mesh sizes 100 and 200. This indicates that there is a notable difference in effectiveness when synthesizing $TiO_2$, with the nanopowder producing a more effective coating.

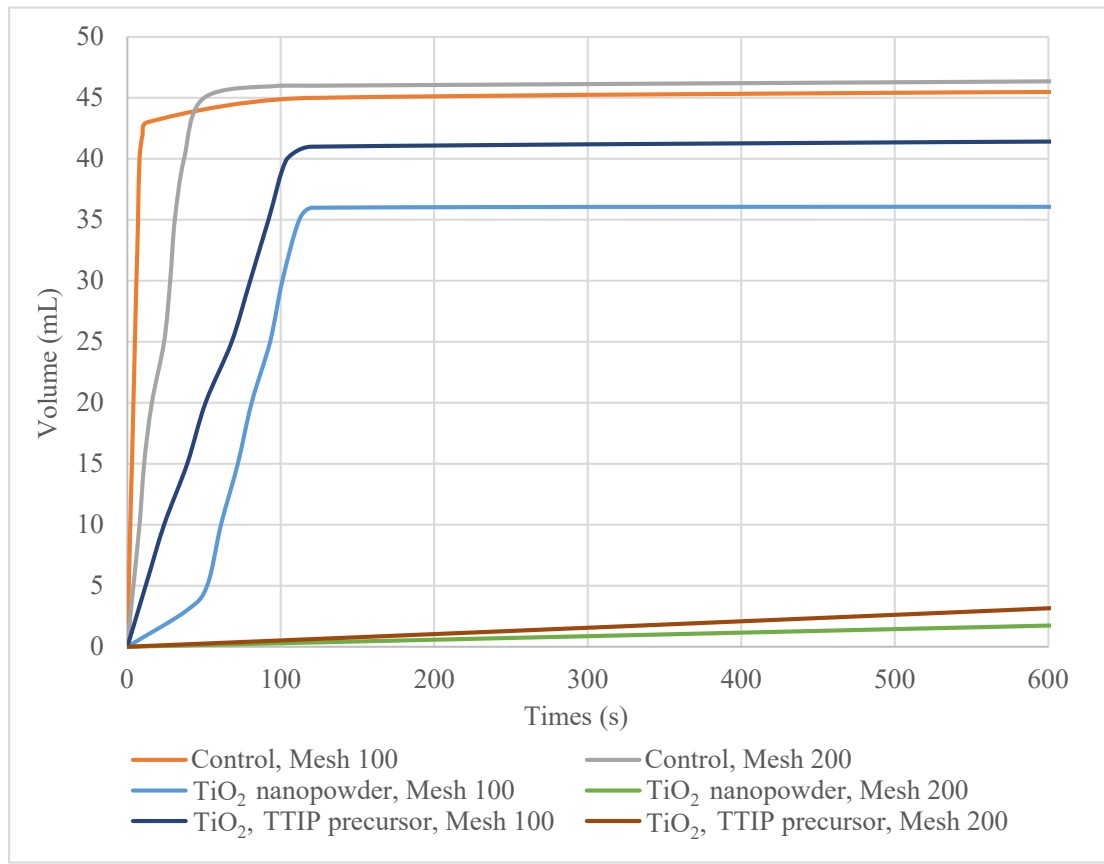

**Figure 8.** Volume of oil filtrates as a function of time.

The results also demonstrate that having a smaller aperture diameter drastically improves oil retention efficiency. Stainless-steel mesh size 200 allows for more contact area between the oleophobic coating and the oil due to its smaller aperture diameter. With larger aperture diameters, such as mesh sizes 80 and 100, applying the $TiO_2$ coating only slightly improves the filtration performance due to less contact area being present. However, using too small of an aperture diameter, in this case mesh size 400, risks clogging the mesh pores, given the current coating procedure.

Figure 9 shows the oil retention efficiency of filters of different aperture sizes and coating methods. The coated mesh 200 filters showed a substantial enhancement compared to the mesh 100 filters. Compared to the control, the coated mesh 200 also had an 82%–84% improvement in efficiency. The $TiO_2$ coating from nanopowder demonstrated slightly better results compared to the one made from the TTIP precursor solution. In the video demonstration provided in the Supplementary Materials, a similar conclusion is seen. In

the video, mesh 200 with the TiO$_2$ nanopowder showed an almost perfect separation of oil and water. In comparison to a similar study regarding the use of amorphous TiO$_2$ [18], the oil retention rate of both coated filters in this work shows a clear improvement. Given a 300 s time frame, the total volume filtered using the coated mesh 200 was below 2 mL. The lower limit of the volume filtered using sonicated coffee filters was approximately 3 mL [18]. Thus, the anatase TiO$_2$-coated stainless-steel filters provide evidence of improved oil blockage from the previous study.

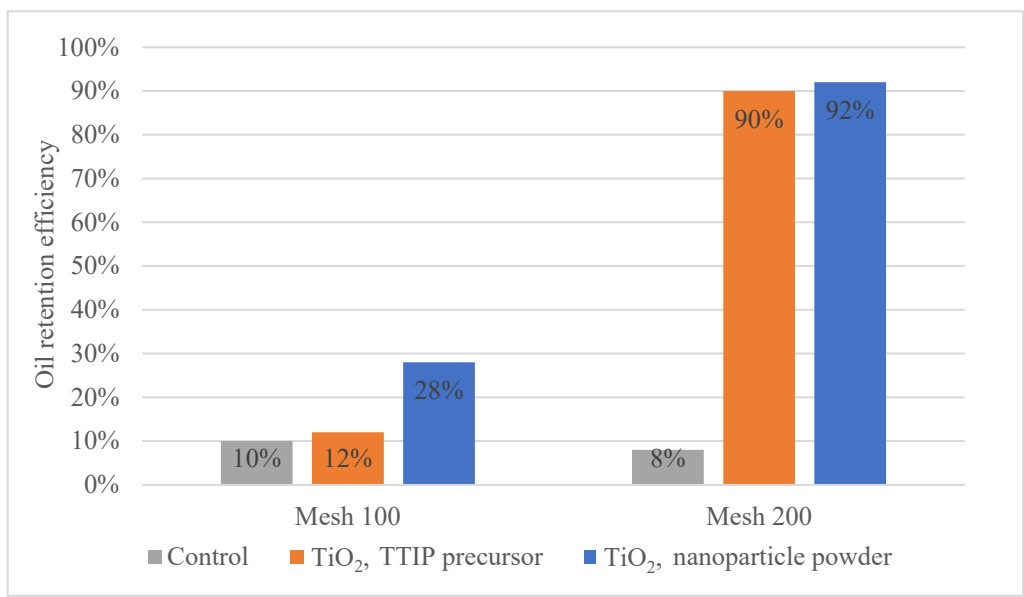

**Figure 9.** Oil retention efficiency of coated filters of varying aperture diameters.

Coating thickness was also a factor to consider in evaluating the performance of the TiO$_2$ nanopowder coating. Thickness was previously demonstrated to be impacted by the number of layers applied to the coating, so multiple coating thickness conditions were tested among varying mesh sizes. Figure 10 demonstrates the general effect of coating thickness on the leakage rate of oil; it confirms the improvement in oil retention due to decreasing aperture size.

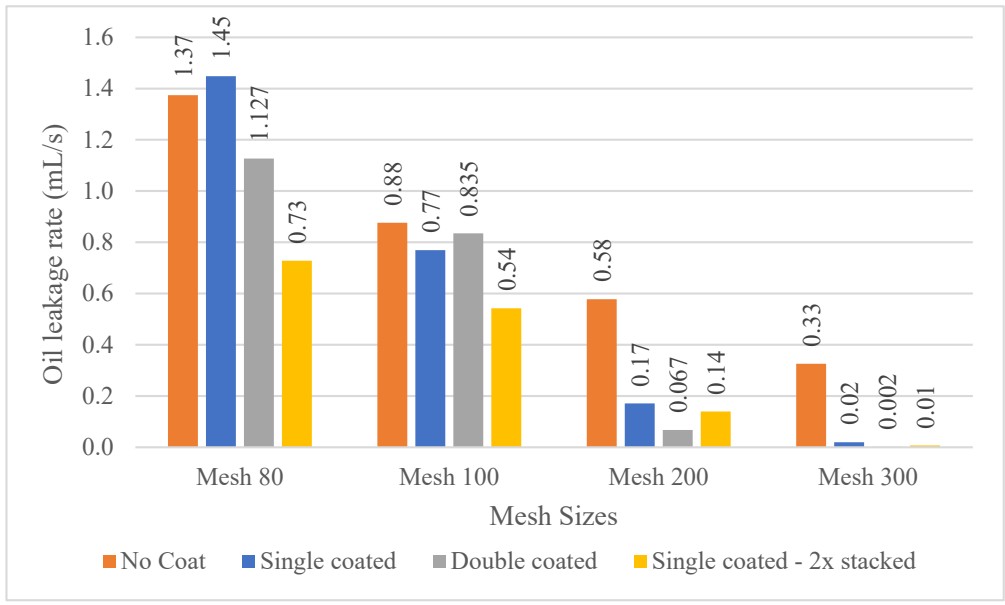

**Figure 10.** Effect of coating thickness on filtration rate of oil.

For each mesh size, general trends are seen. In this case, an improvement to oil and water separation indicates a decrease in the oil leakage rate. Above mesh size 80, most notably for mesh size 200, increasing the thickness of the coating showed clear reductions in the quantity of oil that leaked through the filter. It is worth noting that while the aperture diameter of mesh size 80 may have been too large for the coating thickness to be effective, layering two single-coated filters atop one another provided an improvement to the blockage of oil. Relative to the control, the oil retention of mesh size 200 improved by 71% for single-layer coatings and by 88% for two-layer coatings. This could have originated from the maximization of the filter's oleophobic qualities due to the two factors: a thicker coating and a smaller aperture diameter. These further filtration improvements can also be seen with mesh size 300 compared to mesh size 200, with a 93% and 99% improvement to single- and double-layer coatings, respectively. As aperture diameters decreased, the benefit of using two single-coated filters dwindled, and a clear benefit to using smaller aperture diameters could be seen. It is important to note that, although there was a significant improvement going from mesh 200 to mesh 300, coated mesh 300 took a considerable amount of time to separate oil and water.

### 3.3. Wettability of Oil to Different Oil Filters

Filter wettability directly correlates to the static contact angles measured. Figure 11 displays the contact angles calculated using the ImageJ analysis software version 1.46. It was observed that the contact angle of the $TiO_2$ nanopowder coating decreased from 102° to 93°. While the decrease in contact angle was not predicted, all four samples displayed contact angles greater than 90°. This means that each of the conditions exhibited oleophobic properties. Between the TTIP samples, the mesh 100 and mesh 200 conditions were consistent between contact angle measurements—approximately 100° for the coated mesh 200 and 101° for the coated mesh 100.

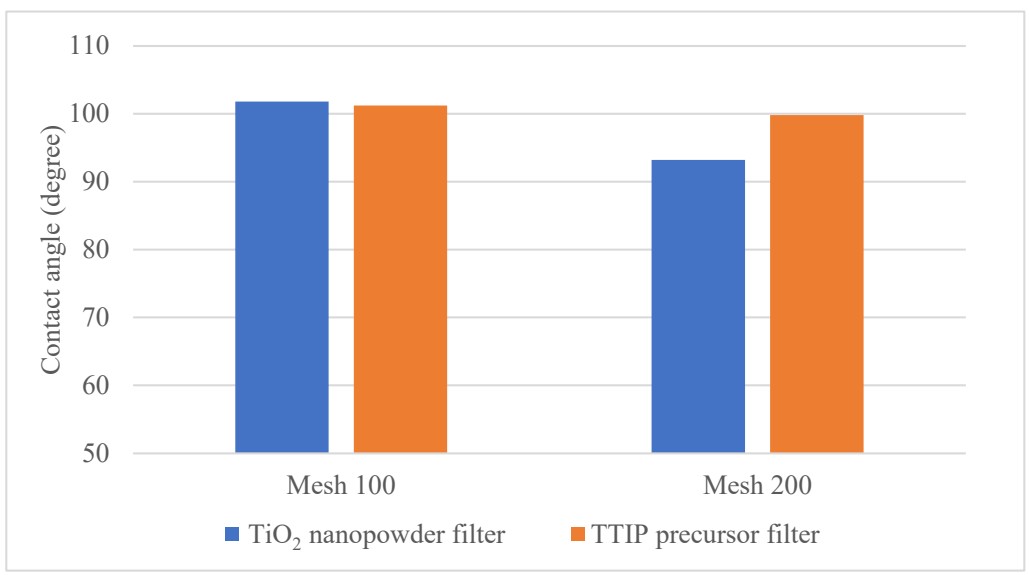

**Figure 11.** Static contact angle test of varying aperture diameter.

The $TiO_2$ nanopowder-coated mesh 200 showed a decrease in contact angle compared to the one based on the TTIP precursor. This may have been because the paste-like substance derived from the TTIP was thicker than the $TiO_2$ nanopowder–acetic acid mixture. A thicker paste has a higher likelihood to block pores. This consequently improves oil leakage rates and contact angle measurements. While this result seems to indicate that a filter with a TTIP-derived coating would provide more efficient filtration, the nanopowder-coated filter provided the best oil/water separation, as indicated from the video demonstration. Regarding a hydrophilicity test, the water droplets were dispersed across the coated

meshes almost instantaneously. This suggests a low contact angle with water, equating to high hydrophilicity.

While both coatings resulted in oleophobic and hydrophilic properties, it was previously concluded that the nanopowder coating outperformed the one from the TTIP precursor during the filtration test. Figure 12 displays an examination of the effect of coating thickness on the effective contact angle of the coating. A clear trend is shown when viewing the Theta average bars in Figure 12. The average measured contact angle increased due to increasing coating thickness as well as decreasing aperture diameter. This correlates with the improved filtration results previously discussed, as increasing contact angle between oil and coated filter indicates increased oleophobicity.

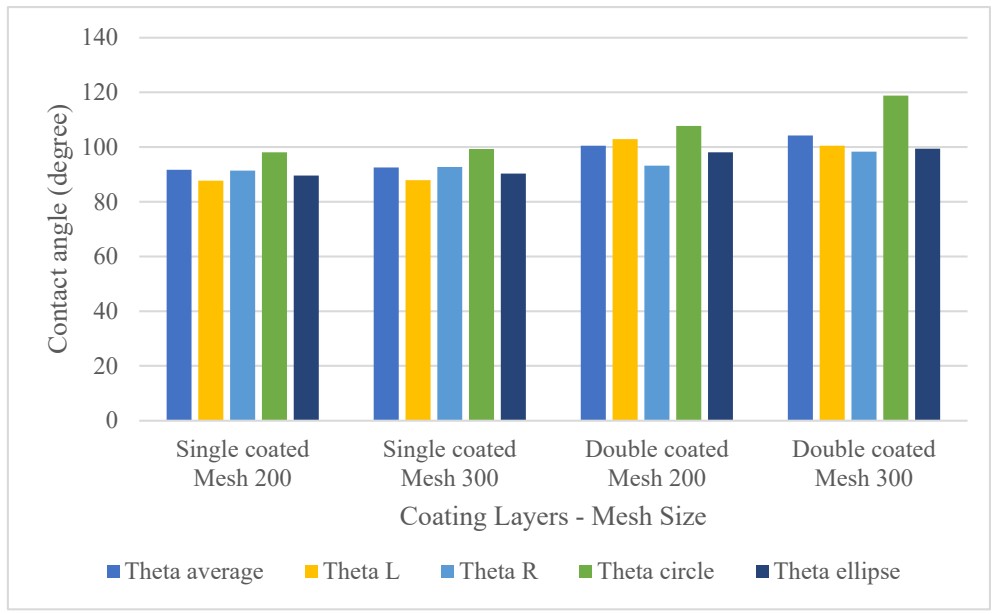

**Figure 12.** Static contact angle measurements of varying coating layers and aperture sizes.

Further improvements to contact angle tests may involve use of the Washburn method via a force tensiometer or use of the Pendant Drop method with the Young–Laplace fit in drop shape analysis.

## 4. Conclusions

The $TiO_2$ coatings derived from the TTIP precursor and $TiO_2$ nanopowder both showed significant improvement in oil and water filtration capabilities for smaller-diameter substrates, but they demonstrated a clogging effect when substrates smaller than 60 microns were used. Applying a thin layer of coating via the spray coating technique helped minimize this issue for mesh sizes 300 and below. The SEM results clearly displayed the applied coating with varying thicknesses, and EDS analysis helped to verify the presence of $TiO_2$ in the coating. XRD results confirmed the presence of anatase-phase $TiO_2$ in the nanopowder coatings following the sintering procedure. Increasing the coating thickness had more notable impacts on the blockage of oil than decreasing aperture diameter. Relative to the controls, the two-layer coatings improved the filtration rate by 88% and 99% for mesh sizes 200 and 300, respectively. Comparing various $TiO_2$-coated meshes with different aperture sizes, the most efficient blockage of oil passage was performed by mesh 200 with the $TiO_2$ nanoparticle coating, closely followed by the $TiO_2$ applied with the TTIP precursor, at 92% and 90% efficiency, respectively.

Cracking of the coating made from the nanopowder was observed in the filtration experiments. It is believed that a substrate with a lower thermal expansion coefficient (e.g., alumina foam) would cause less stress on the coating following large temperature changes via sintering.

**Supplementary Materials:** The supporting information can be downloaded at: https://www.youtube.com/watch?v=WaKF40dOM7k (accessed on 26 May 2023).

**Author Contributions:** Conceptualization, M.L.; methodology, J.S., B.T., D.F., R.M., R.P. and M.L.; formal analysis, J.S., B.T., D.F., R.M. and R.P.; investigation, J.S., B.T., D.F., R.M. and R.P.; resources, M.L.; data curation, J.S., B.T., D.F., R.M. and R.P.; writing—original draft preparation, J.S., B.T., D.F., R.M. and R.P.; writing—review and editing, M.L. and Y.X.G.; supervision, M.L.; project administration, M.L.; funding acquisition, M.L. All authors have read and agreed to the published version of the manuscript.

**Funding:** This work is partially funded by the California State University Council on Ocean Affairs, Science & Technology (COAST-GDP-2023-004) and California Governor's Office of Planning and Research (OPR 19176). The SEM images were made possible through the NSF MRI grant DMR-1429674.

**Institutional Review Board Statement:** Not applicable.

**Informed Consent Statement:** Not applicable.

**Data Availability Statement:** Data are contained within the article.

**Acknowledgments:** The authors would like to thank Hunter Ross and Noor Halabi for their assistance in gathering general information and contact angle data, Jonathan B. Puthoff for his advice in making the nanopowder coating, and Diego Ochoa for his assistance with SEM/EDS and XRD characterization.

**Conflicts of Interest:** The authors declare no conflict of interest. No funders were involved in the design of the study in the collection, analyses, or interpretation, of data; in the writing of the manuscript, or in the decision to publish the results.

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
