# Peer review of "Fabrication and Characterization of TiO2 Coatings on 304 Stainless-Steel Substrate for Efficient Oil/Water Separation"

_coatings, doi:10.3390/coatings13111920_

Round 1
Reviewer 1 Report
Comments and Suggestions for Authors
In the link of the video p YouTube 23 link: https://www.youtube.com/watch?v=WaKF40dOM7k it is not visible where the TiO2 coating was placed, nor is it possible to visualize where the nanoparticles were added.
In the results there is no evidence that the matter is TiO2, it is necessary to add the X-ray diffraction characterization.
The thicknesses of the TiO2 coatings are not shown.
In the contact angle test, the angle measurements are not shown; they must be added.
Expand the conclusions.
Reviewer 2 Report
Comments and Suggestions for Authors
Jose Sico and coauthors have developed a TiO2 coatings on a 304 stainless steel substrate for efficient Oil/Water separation. This work is interesting and will benefit many research fields. However, before publication on Coatings, authors should minorly revise the manuscript and added some necessary experiments/characterizations:
1. The sustainability of this coating on the 304 stainless-steel should be further characterized, including SEM images for the morphology/microstructure of the coating and EDS (Energy Dispersive Spectrometer) mappings for the elements of this coating.
2. The sustainability of this coating should be evaluated, such as in different environmental temperatures, under different deformation of this TiO2-coated stainless-steel, and so on.
3. TiO2 Nanopowder should be further characterized (such as SEM images) to confirm the micro-size and type of the TiO2.
4. Authers should be evaluated the coating thickness dependence on the Oil/Water separation Efficiency.
5. Some of Figures (such as Figure 6) are not clear which should be revised.
Comments on the Quality of English LanguageThe English expression of this manuscript is good with only few errors.
Reviewer 3 Report
Comments and Suggestions for Authors
Decision:
Major revision
Comments
The authors have reported the Thermal Properties of TiO2 Nanoparticles Treated Transformer Oil and Coconut Oil. The article has several technical drawbacks and needs major revision especially in the methods section. The writers are encouraged to address the following topics defined below in order to enhance the scientific rigor of their work.
1. In the abstract I suggest removing the first 3 lines.
2. The author should replace Figure 1 with a flowchart and use some pictures of the techniques used.
3. Figures 1 and 2 need more explanation and what difference the author found in both the samples.?
4. Have you measured the thickness of coated TiO2 film?
5. I suggest adding some surface morphological images like AFM or SEM before and after TiO2 film
6. How did the author prepare the TiO2 nano powder coating? Is it purchased powder, or do you synthesize nano powder? Need some evidence to show its nano powder?
7. It's not clear which sample has better performance author should add a table to highlight the results of each sample.
8. Last there are several grammatical errors. A proofread is required.
Comments on the Quality of English Languageseveral grammatical errors. A proofread is required.
Reviewer 4 Report
Comments and Suggestions for Authors
This paper reports the Fabrication and Characterization of TiO2 Coatings on 304 Stainless Steel Substrate for Efficient Oil/Water Separation. The application of the synthesized materials is good and these analyses are reasonable. However, lack of characterization. Hence, authors should address the following comments for its acceptance.
1. The abstract can be improved with some numerical results drawn from the studies to attract the reader’s attention.
2. How is the synthesis method/work different or better than those reported earlier? Author should highlight this in the introduction part.
3. Materials characterizations are missing. Author needs to confirm the prepared materials in terms of structure and elemental composition.
4. What about the reusability of the prepared materials towards oil/water separation?
5. In order to show the superiority of the current materials, comparisons over the other related materials reported in the past literatures are necessary. Oil/water separation performances of the current materials have to be compared with those of the other materials and reasons for performance improvements have to be discussed.
6. There are some mistakes in the manuscript. The authors need to double-check the whole manuscript to get rid of some syntax and format errors.
Comments on the Quality of English LanguageMinor spell chek is required.
Round 2
Reviewer 1 Report
Comments and Suggestions for Authors
The authors have already made the requested changes, therefore I suggest their publication.
Reviewer 3 Report
Comments and Suggestions for Authors
The Author responded to all my comments and i agree to publish this paper
Comments on the Quality of English LanguageMinor check spelling is required
Reviewer 4 Report
Comments and Suggestions for Authors
The authors have moderately addressed the issues raised by the reviewers. Hence, the revised version of the manuscript may acceptable to the journal standard.